

# Normalized effect size (NES): a novel feature selection model for Urdu fake news classification

Muhammad Wasim[1], Sehrish Munawar Cheema[2] and Ivan Miguel Pires[3,4]

[1] Department of Computer Science, University of Management & Technology, Sialkot Campus, Sialkot, Pakistan
[2] Department of Computer Science, University of Management and Technology, Lahore, Pakistan
[3] Instituto de Telecomunicações, Covilhã, Portugal
[4] Escola Superior de Tecnologia e Gestão de Água, Universidade de Aveiro, Água, Portugal

## ABSTRACT

Social media has become an essential source of news for everyday users. However, the rise of fake news on social media has made it more difficult for users to trust the information on these platforms. Most research studies focus on fake news detection in the English language, and only a limited number of studies deal with fake news in resource-poor languages such as Urdu. This article proposes a globally weighted term selection approach named normalized effect size (NES) to select highly discriminative features for Urdu fake news classification. The proposed model is based on the traditional inverse document frequency (TF-IDF) weighting measure. TF-IDF transforms the textual data into a weighted term-document matrix and is usually prone to the curse of dimensionality. Our novel statistical model filters the most discriminative terms to reduce the data's dimensionality and improve classification accuracy. We compare the proposed approach with the seven well-known feature selection and ranking techniques, namely normalized difference measure (NDM), bi-normal separation (BNS), odds ratio (OR), GINI, distinguished feature selector (DFS), information gain (IG), and Chi square (Chi). Our ensemble-based approach achieves high performance on two benchmark datasets, BET and UFN, achieving an accuracy of 88% and 90%, respectively.

## INTRODUCTION

The term "fake news" represents the stories that are intentionally and undeniably bogus and intended to control individuals' views of genuine realities, events, explanations, and occasions (*Miro-Llinares & Aguerri, 2023*; *Choudhury & Acharjee, 2023*). Fake news covers deception and false data or disinformation deliberately spread to delude individuals (*Ruffo et al., 2023*). It is all about the information projected as news being misleading as it is based on demonstrably incorrect facts and events that never occurred (*Monsees, 2023*; *Cantarella, Fraccaroli & Volpe, 2023*). The internet, social media, and news media provide

Corresponding authors
Sehrish Munawar Cheema,
sehrish.munawar@umt.edu.pk
Ivan Miguel Pires, impires@it.ubi.pt

a platform for millions of users to access up-to-date information and perform social interactions (*Cheng, Ge & Cosco, 2023*; *Longo, 2023*; *Lytos et al., 2019*; *Rodríguez-Ferrándiz, 2023*). The data significantly impacts their opinions and choices for different aspects of their lives (*Xing et al., 2022*; *Kozitsin, 2023*; *Vuong et al., 2019*). Studies have shown that a viral story has an echo chamber effect, and the user is more inclined to have a favorable opinion about it (*Robertson et al., 2023*; *Scheibenzuber et al., 2023*; *González-Bailón & Lelkes, 2023*). Manually differentiating between a real and fake viral story is becoming challenging with the ever-increasing amount of online data (*Khan, Michalas & Akhunzada, 2021*). There are a lot of examples of fake news that we can see throughout history impacting societal values and norms, changing opinions on critical issues, and redefining truths, facts, and beliefs (*Farago, Kreko & Orosz, 2023*; *Olan et al., 2022*). As unreliable and fake information has significant social and economic consequences for society (*Aïmeur, Amri & Brassard, 2023*), it is essential to automatically distinguish between real and fake news (*Buzea, Trausan-Matu & Rebedea, 2022*).

There has been significant research on fake news classification for the English language in the past few years (*Lillie & Middelboe, 2019*; *Shu et al., 2017*; *de Souza et al., 2020*; *Rohera et al., 2022*). However, the work for fake news classification in the Urdu language remains minimal (*Amjad et al., 2020b*; *Amjad, Sidorov & Zhila, 2020*; *Khiljia et al., 2020*) due to the unavailability of a sufficient annotated corpus. Therefore, many challenges must be addressed to solve this problem for the Urdu language. Urdu belongs to the Indo-Aryan group, is written in Arabic Person script, and is the official language of Pakistan. Urdu is spoken by more than 100 million speakers worldwide, but it remains a resource-poor language (*Nazir et al., 2021*; *Ullah et al., 2022*). The datasets for the Urdu language are also small, and deep learning approaches do not perform well on downstream natural language processing tasks such as sentiment analysis and fake news classification (*Rana et al., 2021*). In this regard, one of the foremost concerns is understanding the basis on which an Urdu fake news piece can be classified accurately. It includes extracting appropriate features and ranking discriminative features for Urdu fake news classification.

There are three different paradigms for fake news classification, as shown in Fig. 1: style-based, context-based, and knowledge-based (*Raza & Ding, 2022*). Style-based approaches mainly classify fake news based on deception detection and text categorization (*Kasseropoulos & Tjortjis, 2021*; *Hangloo & Arora, 2021*). Such methods use the news content and extract lexical features to discriminate between real and fake news. Such approaches also require effective feature selection techniques to classify fake news. The context-based paradigm exploits social network analysis to classify fake news (*Donabauer & Kruschwitz, 2023*; *Sivasankari & Vadivu, 2021*). Such approaches use the user's social engagement with the news content and the network of users to identify fake news content. Finally, knowledge-based classification (fact-checking) uses information retrieval techniques, or the semantic web, to detect fake news (*Seddari et al., 2022*; *Pathak & Srihari, 2019*; *Ceron, de Lima-Santos & Quiles, 2021*).

Style-based approaches for fake news classification are based on machine learning and deep learning models. Deep learning models do not perform well on small datasets, and because Urdu is a resource-poor language, no pre-trained models can be fine-tuned for such

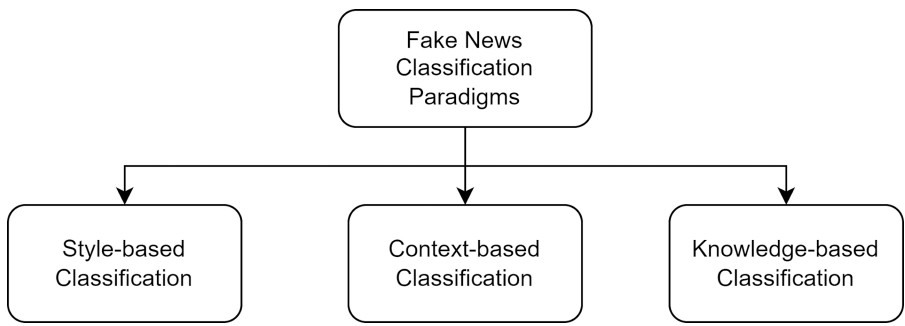

**Figure 1** **Different paradigms for fake news classification (*Potthast et al., 2017*).**

small datasets. Machine learning models are more suitable for resource-poor languages. These models rely on selecting the most relevant features to discriminate between fake and real news (*Rafique et al., 2022*; *Pal, Pranav & Pradhan, 2023*). The first step is to extract the features from the raw text. Secondly, the terms present in the corpus are weighted based on weighting schemes such as the binary vectorizer or TFIDF vectorizer. Finally, different statistical approaches are used for feature vectors and their class labels. After assigning weights to each term, some feature selection measure is used to find and rank the most discriminating terms. In recent studies, ensemble learning approaches outperformed for fake news detection (*Akhter et al., 2021*; *Mahabub, 2020*; *Hakak et al., 2021*; *Fayaz et al., 2022*; *Al-Ash et al., 2019*). Ensemble learning aims to exploit the diversity of base models to increase overall performance by handling multiple error types.

This article proposes a novel feature weighting model named normalized effect size (NES) that is based on a global weighting scheme (TFIDF). We extracted three features from the Urdu news articles: word n-grams, character n-grams, and function n-grams. The top k terms ranked by NES are used to train an ensemble model. The performance of the proposed approach is compared with seven feature selection measures, namely Normalized Difference Measure (NDM), Bi-normal Separation (BNS), Odds Ratio (OR), GINI, Distinguished Feature Selector (DFS), Information Gain (IG) and Chi Square (Chi). Our significant contributions to this research are as follows:

1. Proposal of a robust feature selection approach to discriminate between fake and real news;
2. Comparison of the performance of the proposed feature metric with seven well-known feature selection methods, showing the high performance of our feature metric;
3. Analysis of the performance of the proposed feature selection metric on two benchmark Urdu fake news datasets.

All example articles in the benchmark datasets have two classes assigned to them, *i.e.*, real or fake news. Therefore, our problem is a binary classification problem compared to the multi-class classification problem, in which more than two classes are used for labeling a dataset's examples. The following section briefly reviews the literature on fake news classification.

## LITERATURE REVIEW

Previously, researchers have used feature selection for machine learning on different genres of textual data (*Katakis, Tsoumakas & Vlahavas, 2005*; *Rehman, Javed & Babri, 2017*; *Khan, Alam & Lee, 2021*; *Ramasamy & Meena Kowshalya, 2022*). In this section, we shall focus on feature selection for fake news classification, specifically for Urdu fake news classification. We shall review the feature selection techniques that have been used previously for the same or related tasks and summarize their findings.

Urdu is the national language of Pakistan and the 8th most spoken language globally, with more than 100 million speakers (*Akhter et al., 2020*). It is a South Asian language with limited resources (*Nazir et al., 2021*). A few annotated corpora in a few domains are available for research purposes, compared to English, which is a resource-rich language (*D'Ulizia et al., 2021*). The availability of insufficient linguistic resources like stemmers and annotated corpora makes the research on Urdu fake news classification more challenging and inspiring. Labeling a news article as "fake" or "legitimate" requires experts' opinions and is time-consuming. Also, hiring experts for each related domain is costly. In *Amjad et al. (2020b)*, *Amjad, Sidorov & Zhila (2020)*, *Amjad et al. (2020a)* and *Amjad et al. (2022)*, the authors proposed an annotated fake news corpus with a few hundred news articles. Their experimental results reveal the poor performance of machine learning models. Deep Learning models perform poorly due to the small corpus available for Urdu fake news.

Ensemble learning techniques boost the efficiency of individual machine learning models by aggregating the predictions in a way, also called base learners, base models, and base predictors (*Sagi & Rokach, 2018*). *Mahabub (2020)* applied eleven machine learning classifiers on a fake news corpus, including neural network-based models. Three of eleven machine learning models were selected to ensemble a voting model. Ensemble soft voting results reflect better performance than other models. To detect fake reviews, two ensemble learning approaches, bagging and boosting, were applied with SVM and MLP-based learners, and their research findings reflect that boosting with MLP outperforms the others (*Singh & Selva, 2023b*; *Singh & Selva, 2023a*; *Gutierrez-Espinoza et al., 2020*). Numerous ways are used to achieve this, such as re-sampling the corpus, heterogeneous models, homogeneous models with diverse parameters, and using various methods to combine the predictions of base models (*Kunapuli, 2023*).

Machine learning models are applied for fake news detection and classification tasks for languages such as English, Portuguese, Urdu, Arabic, Spanish, and Slavic (*Lahby et al., 2022*; *Nirav Shah & Ganatra, 2022*; *Ahmed et al., 2021*). Less effort has been made to explore ensemble learning techniques to fake news classification compared to machine learning methods (*Capuano et al., 2023*; *Chiche & Yitagesu, 2022*).

*Posadas-Durán et al. (2019)* defined three categories to detect fake news: knowledge-based, context-based, and style-based. They used bag-of-words (BOW), POS tags, and n-gram features. In normalization, they removed tags such as the editor's emails or phone numbers. Their experiments showed that the random forest outperformed with the highest accuracy of 76.94% using BOW, POS, and n-grams. Their work was primarily focused on the Spanish language.

*Reis et al. (2019)* used PolitiFact, Channel 4, and Snoops to gather fake news articles. They used linguistic inquiry word count (LIWC) and could attain an accuracy of 60%. Similarly, in another similar study (*Ahmed, Traore & Saad, 2018*), the authors used two feature weighting techniques, TF and TF-IDF, along with N-gram features. Their extracted features with the linear SVM classification model outperformed the baseline with an accuracy of 90% by using Bigram and 10,000 features.

*Krešňáková, Sarnovskỳ & Butka (2019)* used preprocessing techniques such as stopwords, punctuation removal, and word2vec word embedding to represent the features. They conducted experiments with four classification models: feedforward neural networks, CNNs with one convolutional layer, CNNs with more convolutional layers, and LSTMs. They used only textual data to train the model and achieved the highest F1 score of 97.52% by using CNN mode. Using the CNN model, they also used the title and text to train the model and achieved an F1-score of 93.32%.

*Bajaj (2017)* collected a fake news dataset from Kaggle. They used different classification models: logistic regression (LR), feedforward neural networks, recurrent neural networks (RNN), long short-term memory (LSTM), gated recurrent units (GRU), bidirectional RNN with LSTM, a convolutional neural network with max-pooling, and an attention-augmented convolutional neural network (CNN). In this model comparison, GRU achieved the highest F1-score of 84%, and the performance of CNN was deficient (6%). Similarly, *Saikh et al. (2020)* collected the dataset in six domains and analyzed it. They used hand-crafted linguistic features and support vector machines. They achieved accuracies of 74% and 76% in the AMT and Celebrity News datasets, respectively. Also, they solved this problem using deep learning approaches. The first model was Bi-directional Gated Recurrent Unit (BiGRU), and the second was embedded from the language model (ELMo) and got accuracies of 54% and 68%, respectively.

*Granik & Mesyura (2017)* worked on unsolicited messages (spam) and fake news. Both types of messages have grammatical mistakes and false content. Fake news and spam both have some similarities. The set of words in one spam article is also in fake news and other spam articles. In this paper, the source dataset was BuzzFeed News. This dataset was contained in Facebook news posts. They collected data from three news pages (ABC News, Politico, and CNN). The dataset contained 2,282 posts. After cleaning the data, they obtained 1,771 articles. This data was classified into three subsets (training, validation, and testing). For true probability, the threshold value was [0.5; 0.9]. The unconditional probability of the news article was 59%, and the true probability of the threshold was 80%. A total number of fake news articles in the dataset contained 46 news articles, and 333 news articles were successfully classified with 71.73% accuracy. From a total of 927 articles, 699 were successfully classified. The accuracy of true news articles was slightly better than that of fake news articles. They used 2,000 articles, and this dataset was minimal to improve performance. The performance would also improve by using stemming and removing stop words. The total number of true news articles in the dataset contained 881 and 666 news articles successfully classified with 75.99% accuracy.

*Liu & Wu (2018)* collected three datasets (Weibo, Twitter15, and Twitter16) from Chinese and US social media sites. In the Weibo dataset, stories have binary labels, *i.e.,* fake

and true. On the other hand, in the Twitter15 and Twitter16 datasets, stories contained four labels (fake, true, unverified, and debunked). The fake news detection modal had four components: propagation path construction and transformation, RNN-based propagation path representation, CNN-based propagation path representation, and propagation path classification. They used 75% of the collected dataset for training and 25% of the dataset for testing. They gathered user information from their profiles. Furthermore, they found eight common characteristics between Weibo and Twitter and used stochastic gradient descent to train the model over 200 epochs. They used three proposed models: propagation path classification (PPC) (PPC_RNN, PPC_CNN, PPC_RNN+CNN). They compared their model with the baseline comparison model (DTC, SVM-RBF, SVM-TS, DTR, GRU, RFC, and PTK). The performance of the PPC_RNN+CNN model was the best among all. The accuracy of Twitter15 was 84.2%, Twitter16 was 86.3%, and Weibo was 92.1%.

*Vogel & Meghana (2020)* worked in two languages (English and Spanish). They used the n-gram feature and support vector machine (SVM) for English and logistic regression for Spanish. They used the PAN 2020 dataset, which has 300 English and 300 Spanish Twitter accounts, and each user had 100 tweets. Likewise, they removed stop words by using the NLTK library. They used 70% of the data for training and 30% for testing. The accuracy using SVM and the TF-IDF char n-gram feature in English was 73%. Also, the accuracy using logistic regression and the TF-IDF char n-gram feature in Spanish was 79%.

*Ahmed, Traore & Saad (2018)* used n-gram and machine learning techniques. They used two feature extraction techniques (term frequency-inverted document (TF-IDF) and n-gram) and six classification techniques (Stochastic Gradient Descent (SGD), Decision Tree (DT), Linear Support Vector Machine (LSVM), Logistic Regression (LR), Support Vector Machine (SVM) and K-nearest neighbor (KNN)). The TF-IDF feature and LSVM classifier achieved the highest accuracy of 92%. The dataset contained 25,200 articles, with an equal number of real and fake documents. They used n-gram features ranging from 1 to 4. In all the non-linear classifiers, DT achieved the highest accuracy of 89%. The linear base classifiers (LR, LSVM, and SDG) performed better than the non-linear classifiers. The performance of the model decreased as the number of n-grams increased. KNN and SVM achieved the lowest accuracy of 47.2%. The performance of SVM with a linear kernel attained an accuracy of 71%.

*Monti et al. (2019)* used deep geometric learning for fake news detection. The author collected datasets from Snopes, Politifact, and BuzzFeed. They extracted features into four categories: user profile, activity, network and spreading, and content. They used four layers of CNN graphs and two convolutional layers to predict the probability of fake and real articles. Furthermore, they used scaled exponential linear units (SELU) in their network. The dataset was split into training (677), testing (226), and validation (226) for URL classification. They used the same pattern to split data, like URL classification for cascade classification. The ROC AUC was 92.70 ± 1.80% URL-wise classification and 88.30 ± 2.74% cascade-wise classification. For URL settings, they split the data into 80% of URLs for training and 20% for testing. In the ablation experiment for both settings, two features (user profile and network spreading) had high importance, with nearly 90% ROC AUC. There was a time t for the first tweet, and t was 0 to 24 h. Each value of the

t model was trained separately. They also used five cross-validations to reduce bias. The method improved performance with cascade duration. There was different behavior due to different characteristics of cascades and URLs. The highest performance was 92.7% ROC AUC.

Only a handful of researchers have worked on fake news in the Urdu language. *Amjad et al. (2020b)* manually collected and verified the datasets. Their dataset contained 500 real and 400 fake news articles. They used raw frequency, a binary weighting scheme, normalized frequency, log entropy, and TF-IDF. They got a good performance with the combinations of character-word 2-grams and 1-grams, reporting a F1-score of 87% for fake news and a F1-score of 90% for legit news using the AdaBoost classifier. In a different study by *Amjad, Sidorov & Zhila (2020)*, the researchers developed a new dataset using machine translation (MT). The model's performance on the new dataset was not satisfactory compared to the original. Binary weighting schemes were used for feature normalization (TF-IDF) and log-entropy (decrease classification performance). They used Support Vector Machines (SVM) and AdaBoost classifiers. Then they used the dataset of *Amjad et al. (2020b)*, using the Original Urdu dataset combination of character and word Unigrams, and achieved excellent results with an F1-score of 84% for fake and a ROC-AUC score of 94%.

*Khiljia et al. (2020)* utilized generalized autoregressive-based model techniques on a dataset of 350 real and 288 fake articles. The data was preprocessed using the UrduHack library, removing mobile numbers, email IDs, URLs, and extra spaces. The XLNet pre-trained model was used for training, but BERT was used for fine-tuning. The study achieved a F1 macro score of 83.70%.

The CharCNN-RoBETa model was proposed by *Lina, Fua & Jianga (2020)*, using a dataset of 500 real and 400 fake articles. The model represented sentences as word and character embeddings by concatenating vectors and applying softmax for prediction. Five-fold cross-validation was used to achieve high accuracy. The model achieved a F1-score of 99.99% with the RoBERTa+pretrain model, 91.18% with the RoBERTa+pretrain+label smoothing model, 91.25% with the RoBERTa+charcnn+pretrain model, and 91.41% with the RoBERTa+charcnn+pretrain model.

Table 1 presents a summary of the comparison of previous work and the features used for Fake News Classification, where it is verified that the word n-gram.

## PROPOSED METHODOLOGY

Our proposed methodology for Urdu fake news classification follows four steps, where the first step involves feature extraction from the documents. Next, we apply different feature weighting schemes to the feature vectors. Once weights are assigned, we use state-of-the-art feature selection techniques to select highly discriminative features and compare them with our proposed feature selection metric. Finally, the classification model (AdaBoost) is trained based on these features. This methodology was implemented in *Wasim (2023)*, and Fig. 2 shows the complete process with the details of each component in the following subsections.

Wasim et al. (2023), *PeerJ Comput. Sci.*, DOI 10.7717/peerj-cs.1612

**Table 1 Comparison of previous work and the features used for fake news classification.**

| No. | Paper | Features | Models | Performance |
|---|---|---|---|---|
| 1 | *Amjad et al. (2020b)* | TF-IDF, log-entropy, character n-grams, word n-grams | AdaBoost | 84% F1 Fake, 94% ROC-AUC scores, lower ROC-AUC of 93%. |
| 2 | *Amjad, Sidorov & Zhila (2020)* | Word n-gram, char n-grams, and functional n-grams | AdaBoost Classifier's | Accuracy 87%, F1 Fake 91% |
| 3 | *Monti et al. (2019)* | User profile, User activity, Network and spreading, Content | Convolutional Neural Network (CNN) | 92.7% ROC AUC |
| 4 | *Humayoun (2022)* | Word n-gram, character n-gram | Support Vector Machine, CNN Embeddings | F1 macro 66%, Accuracy 72% |
| 5 | *Rafique et al. (2022)* | TF-IDF, BoW, Character N-gram, word N-gram | NF, LR, SVC, GB, PA, Multinomial NB | Accuracy 95% |
| 6 | *Amjad et al. (2020a)* | Character bi-gram, MUCS, BoW, Random | BERT 4EVER, Logistic Regression | Accuracy 90% |
| 7 | *Amjad et al. (2022)* | TF-IDF, count-based BoW, word vector embeddings | SVM, BERT, RoBERta | F1-macro 67%, Accuracy 75% |
| 8 | *Kalra et al. (2022)* | N/A | Ensemble Learning, ROBERTA, ALBERT, Multilingual Bert, xlm-RoBERTa | Accuracy 59% |
| 9 | *Salahuddin & Wasim (2022)* | TF-IDF | Logistic Regression | F1 Score 72% |
| 10 | *Akhter et al. (2021)* | BoW, IG | SVM, Decision Tree, Naive Bayes | BA 81.6%, AUC 81.5%, MAE 23.5% |

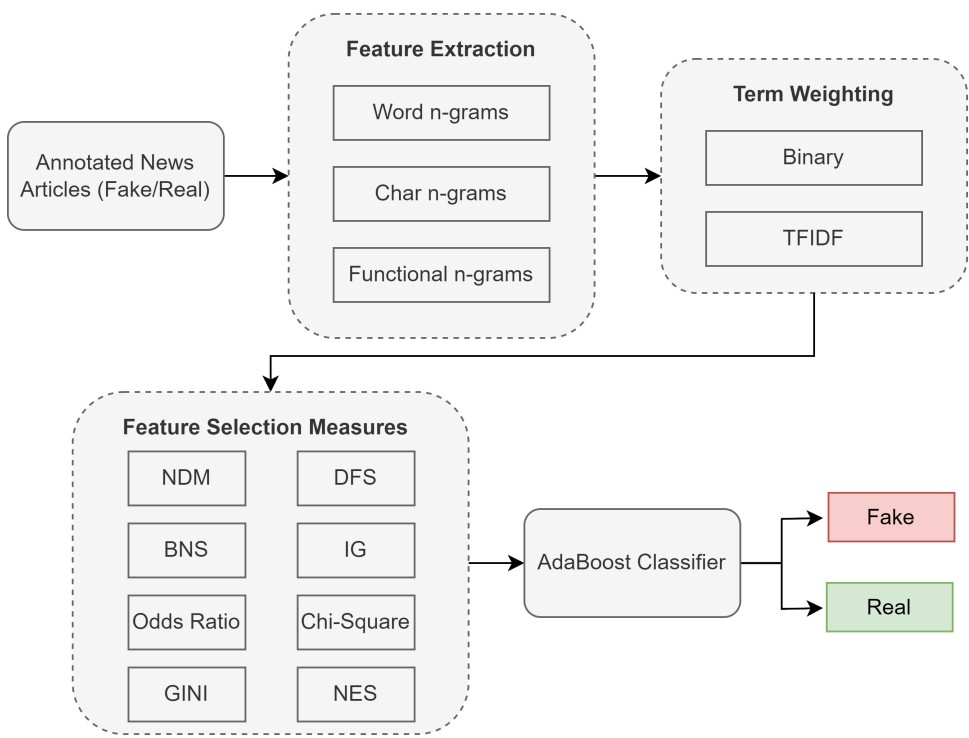

**Figure 2** **Proposed methodology with different feature selection techniques for Urdu fake news classification.**

## Feature extraction

We extracted three types of features from our datasets: word n-grams, character n-grams, and functional n-grams.

**Word n-grams:** The first feature we extracted from the document is word unigrams. We use the unigram feature, as we observed that these features have superior performance and low sparsity compared to higher-order n-grams for fake news classification.

**Character n-grams:** The second feature extracted from the document is character n-grams. The purpose of this feature is to capture the syntactic and morphological elements present in the document. We extracted 2-gram sequences of characters for the character n-gram feature.

**Functional word n-grams:** The third feature is functional word n-grams. We extract these features as previous studies on fake news classification have shown improved performance using functional words. The functional words include determiners, prepositions, articles, and auxiliary verbs. The functional word n-grams is a sequence of these words, omitting the content words, and we use a sequence of 2-gram functional words.

## Term weighting schemes

Term weighting for the extracted features is vital to improving the performance of fake news classification. We perform experiments with two different types of term weighting schemes:

**Binary weighting:** In this weighting scheme, we use binary feature values. If a feature is present one or more times in the document, it is assigned a value of 1. Otherwise, it is assigned the value of zero.

**TFIDF weighting:** TFIDF is a well-known weighting measure used in information retrieval and classification. It is calculated as:

$$TFIDF = TF(t,d) \times log(\frac{N}{df_i}) \tag{1}$$

where TF is the value of a term *t's* presence in document *d*, $df_i$ is the document frequency of term $t_i$, and *N* is the total number of documents in the corpus.

## Feature selection (FS) methods

Feature selection methods play a vital role in machine classification tasks. We provide the necessary details of the state-of-the-art feature selection methods for comparison. We start by defining the true positive (TP), true negative (TN), false positive (FP), and false negative (FN) of a term that forms the basis of all the feature selection measures, where TP is the number of documents in the positive class with the term in them, TN is the number of documents that are not in the positive class and have no term, FP is the number of documents in the negative class with the term in them, and FN is the number of documents that are not in the negative class and have no term.

### *Normalized difference measure (NDM)*

Normalized difference measure (NDM) is a new feature selection technique proposed by *Rehman, Javed & Babri (2017)*. This is based on *tpr* and *fpr* where:

$$tpr = \frac{TP}{N_{pos}} \tag{2}$$

$$fpr = \frac{FP}{N_{neg}} \tag{3}$$

where $N_{pos}$ is the number of positive documents and $N_{neg}$ is the number of negative documents in the corpus. NDM is based on the following principles:

- An important term should have high $|tpr - fpr|$ value.
- One of the tpr or fpr values should be closer to zero.
- If two terms have equal $|tpr - fpr|$ values, then the term having a lower min (tpr, fpr) value should be assigned a higher rank where min is the function to find a minimum of the two values.

Mathematically, NDM is defined as:

$$NDM = \frac{|tpr - fpr|}{min(tpr, fpr)} \tag{4}$$

Note that if the min of *tpr* and *fpr* is zero, a smaller value like 0.01 is used.

### Bi-normal separation (BNS)

*Forman (2003)* proposed bi-normal separation and is defined as:

$$BNS = |F^{-1}(tpr) - F^{-1}(fpr)| \qquad (5)$$

where $F^{-1}$ is the standard Normal distribution's inverse cumulative probability function or z-score. To avoid the undefined value $F^{-1}(0)$, zero is substituted by 0.0005.

### Odds ratio (OR)

The odds ratio depends on the probability of a term's occurrence or whether that term is not present in a document. This feature extraction technique only relies on the probability of the occurrence of terms.

$$OddsRatio = \frac{TPR(1 - FPR)}{(1 - TPR)FPR} = \frac{TP \times TN}{FP \times FN}. \qquad (6)$$

If fp or fn is zero, the denominator will be zero values ($fp * fn = 0$), which is replaced with a small value.

### Gini

The Gini index was initially used to estimate income distribution across a population. It is also used as a feature ranking metric, where it is used to estimate the distribution of an attribute over different classes.

$$GINI = tpr^2 \times \left(\frac{tp}{tp \times fp}\right)^2 + fpr^2 \times \left(\frac{fp}{tp \times fp}\right)^2 \qquad (7)$$

### Distinguished feature selector (DFS)

*Uysal & Gunal (2012)* proposed a distinguished feature selector based on the idea that the terms present in one class only play an important role in discriminating between the two classes.

$$DFS(t) = \sum_{i=1}^{m} \frac{P(c_i|t)}{P(\bar{t}|c_i) + P(t|\bar{c}_i) + 1} \qquad (8)$$

where $P(c_i)$ is probability of ith class and $P(\bar{t}|c_i)$ is probability of absence of term $t$ when class $c_i$ is given.

### Information Gain (IG)

Information gain is widely used to assess the usefulness of features for machine learning. It measures the decrease in entropy when the feature is given *vs.* when the feature is absent. It is defined as:

$$IG(t) = -\sum_{i=1}^{m} P(c_i)logP(c_i) - \left(P(t)\sum_{i=1}^{m}P(c_i|t)logP(c_i|t) + P(\bar{t})\sum_{i=1}^{m}P(c_i|\bar{t})logP(c_i|\bar{t})\right) \qquad (9)$$

where $c_i$ is set of classes in the dataset, $P(c_i)$ is probability of *ith* class and $P(c_i|t)$ is probability of *ith* class when term $t$ is present and $P(c_i|\bar{t})$ is probability of class $c_i$ when term $t$ is absent.

### Chi square (Chi)

Chi-square (Chi) measures the divergence from the expected distribution, assuming that the presence or absence of a term is independent of the class label (*Forman, 2003*). It is defined as:

$$CHI = t(tp, (tp+fp)P_{pos}) + t(fn, (fn+tn)P_{pos}) + t(fp, (tp+fp)P_{neg}) + t(tn, (fn+tn)P_{neg}) \quad (10)$$

where $t(count, expect) = (count - expect)^2/expect$, $P_{pos}$ is the probability of the positive class and $P_{neg}$ is the probability of the negative class.

### Propose feature selection measure: normalized effect size (NES)

According to *Sullivan & Feinn (2012)*, the effect size is used in medical research to determine the magnitude of the difference between two groups. Inspired by their idea, we define normalized effect size (NES) as the absolute mean difference between a term's TF-IDF score distribution in positive and negative classes normalized by the sum of their standard deviation. NES is a global weighting technique for discriminative feature selection, mathematically defined as:

$$NES = \frac{|\mu_+ - \mu_-|}{\sigma_+ + \sigma_-} \quad (11)$$

where:

- $\mu_+$ is the mean of the TFIDF score of a term across all the documents labeled as positive
- $\mu_-$ is the mean of the TFIDF score of a term across all the documents labeled as negative
- $\sigma_+$ is the standard deviation of the TFIDF score of a term across all documents labeled as positive
- $\sigma_-$ is the standard deviation of a TFIDF score of a term across all documents labeled as negative

## Classification model

Previous studies show that ensemble learning approaches, such as bagging and boosting, outperform the other algorithms for Urdu fake news classification. Therefore, we use the adaptive boosting technique, which boosts multiple serial estimators, to improve the classification performance. We experimentally set the hyperparameter values for alpha (learning rate) and the number of estimators. The value of the learning rate hyperparameter was set to 0.1, and the number of estimators for AdaBoost was set to 300.

# EXPERIMENTAL EVALUATION

This section discusses the corpus used to evaluate the proposed methodology for evaluating different feature selection metrics. We use a well-known accuracy measure to analyze and compare feature selection measures. The results and comparison are presented in the last subsection.

## Evaluation corpus

We use two datasets to evaluate the performance of our proposed feature selection measure, namely the Bend the Truth (BET) dataset (*Amjad et al., 2023*) and the Urdu Fake News (UFN) dataset (*Akhter, 2023*).

**Table 2** The number of real and fake news documents present in each category for the BET dataset.

| Category | Real | Fake |
|---|---|---|
| Business | 100 | 50 |
| Health | 100 | 100 |
| Showbiz | 100 | 100 |
| Sports | 100 | 50 |
| Technology | 100 | 100 |
| **Total** | 500 | 400 |

**Table 3** Classification performance on feature ranking metrics with varying number of terms on UFN dataset.

| No. of terms | Feature selection measures | | | | | | | |
|---|---|---|---|---|---|---|---|---|
| | NDM | BNS | OR | Gini | DFS | IG | CS | NES |
| 100 | 0.78 | 0.77 | 0.63 | 0.55 | 0.85 | 0.84 | 0.84 | 0.88 |
| 200 | 0.78 | 0.8 | 0.65 | 0.63 | 0.87 | 0.86 | 0.85 | 0.89 |
| 300 | 0.8 | 0.83 | 0.64 | 0.72 | 0.87 | 0.87 | 0.86 | 0.90 |
| 400 | 0.8 | 0.84 | 0.67 | 0.76 | 0.86 | 0.86 | 0.87 | 0.89 |
| 500 | 0.81 | 0.85 | 0.67 | 0.81 | 0.88 | 0.87 | 0.87 | 0.89 |
| 1,000 | 0.85 | 0.86 | 0.73 | 0.84 | 0.89 | 0.89 | 0.88 | 0.89 |
| 1,500 | 0.86 | 0.87 | 0.74 | 0.87 | 0.89 | 0.88 | 0.89 | 0.89 |
| 2,000 | 0.85 | 0.88 | 0.75 | 0.88 | 0.88 | 0.88 | 0.88 | 0.89 |

The BET dataset comprises five categories: business, health, showbiz, sports, and technology. The corpus contains 900 documents (500 real and 400 fake new documents). The statistics of each category with the number of real and fake news are shown in Table 2.

The Urdu Fake News (UFN) dataset contains 1,032 real and 968 fake news stories, representing 2,000 stories. The dataset does not provide any categories and is just a translation of the fake news English corpus. We use 65% of the data for training and 35% for testing purposes for both datasets.

## Results and comparison

This section presents the performance of different feature selection metrics on the BET and UFN datasets. We compare the proposed method with seven well-known feature selection measures. These feature selection measures are evaluated on different numbers of terms selected, from 50 to 2,000 top terms. The section also compares the results of the proposed method with previous results on the same datasets.

Table 3 shows the results of the performance of the AdaBoost classifier for a different number of terms selected by the feature selection measures in our study compared with the proposed feature selection measure. The proposed measure performs better than all other feature metrics over all the different numbers of terms. The results are also depicted in Fig. 3.

Surprisingly, both binary and TF-IDF scores perform similarly on the UFN dataset, as shown in Fig. 3, attaining an accuracy of 89%. The odds ratio (OR) performance is worst

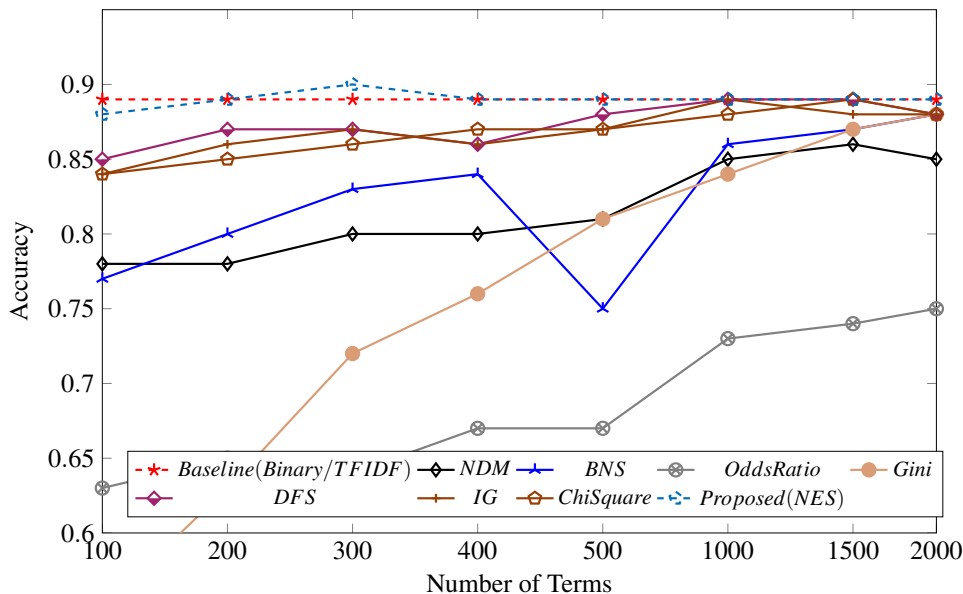

**Figure 3** Feature selection measure applied to the UFN dataset.

**Table 4** Classification performance on feature ranking metrics with varying number of terms on BET dataset.

| No. of terms | Feature selection measures | | | | | | | |
|---|---|---|---|---|---|---|---|---|
| | NDM | BNS | OR | Gini | DFS | IG | CS | NES |
| 100 | 0.69 | 0.85 | 0.74 | 0.51 | 0.88 | 0.88 | 0.88 | 0.87 |
| 200 | 0.81 | 0.78 | 0.8 | 0.8 | 0.86 | 0.84 | 0.85 | 0.88 |
| 300 | 0.82 | 0.76 | 0.76 | 0.81 | 0.85 | 0.85 | 0.85 | 0.86 |
| 400 | 0.83 | 0.73 | 0.78 | 0.84 | 0.87 | 0.83 | 0.86 | 0.86 |
| 500 | 0.85 | 0.77 | 0.79 | 0.86 | 0.87 | 0.87 | 0.87 | 0.87 |
| 1,000 | 0.84 | 0.85 | 0.82 | 0.86 | 0.87 | 0.87 | 0.87 | 0.85 |
| 1,500 | 0.85 | 0.85 | 0.83 | 0.87 | 0.85 | 0.87 | 0.86 | 0.85 |
| 2,000 | 0.84 | 0.84 | 0.84 | 0.85 | 0.85 | 0.87 | 0.87 | 0.85 |

across all different numbers of terms. All the other measures could not perform well in selecting discriminative terms for fake news classification.

Table 4 shows the feature selection metrics results compared with the proposed methodology. It shows that our proposed method performs well for 100, 200, and 300 features, achieving the highest accuracy of 88% with 200 top features. Similar to our observation for the UFN dataset, the binary and TF-IDF weightings achieved identical results, as shown in Fig. 4.

We compare the performance of feature weighting techniques without any feature selection metric and compare their performance when our proposed method is used, as shown in Fig. 5. It shows that performance significantly improves when our proposed feature selection metric is used for the classification task on both datasets. The performance

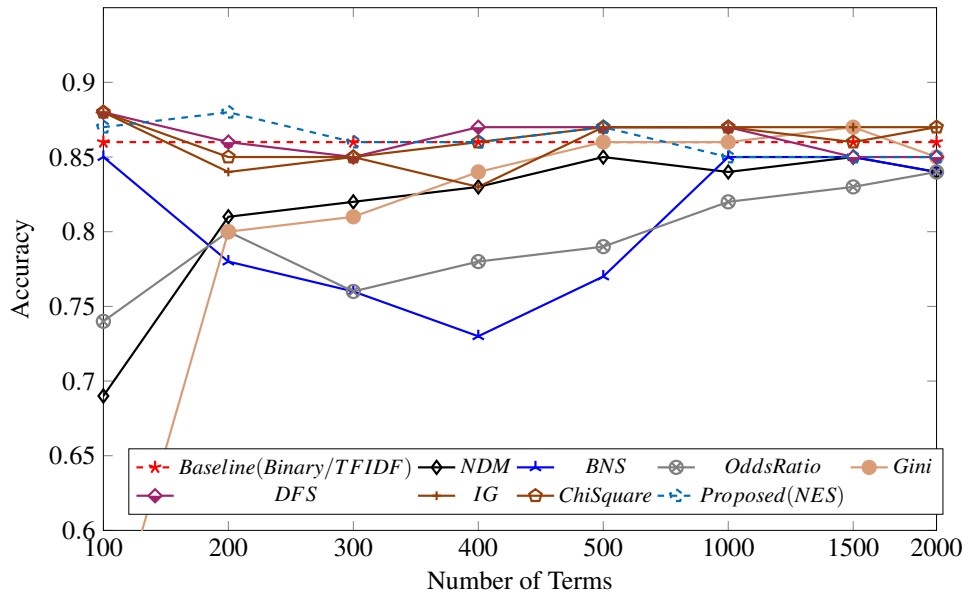

**Figure 4** Feature selection measure applied to the BET dataset.

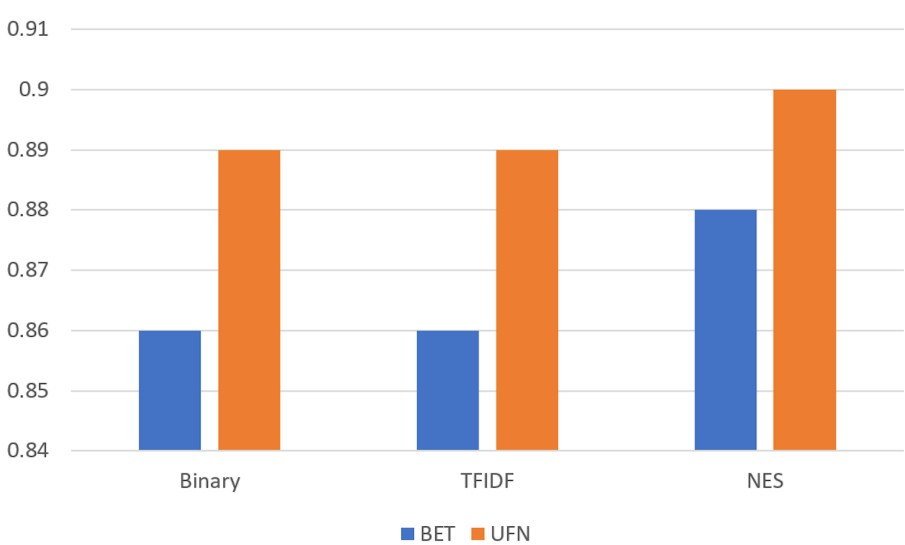

**Figure 5** Comparison on BET and UFN dataset for simple weighting compared with the proposed feature selection metric (NES).

on the BET dataset is low for all weighting measures and the proposed feature selection measure. The reason for this is the small dataset size.

We also compare the performance of the proposed method and the final results with previous state-of-the-art results on the same dataset. Table 5 compares the achieved results to those of prior studies. The results show that the feature selection metric-based methodology performs better on both datasets.

**Table 5  Comparison of the proposed method with the previous studies.**

| Study | BET dataset | UFN dataset |
|---|---|---|
| *Salahuddin & Wasim (2022)* | 72% | NA |
| *Amjad et al. (2020b)* | 83% | NA |
| *Akhter et al. (2020)* | 83% | 89% |
| Proposed approach (NES) | 88% | 90% |

## CONCLUSION

The findings of this study demonstrate the critical need for effective feature selection techniques for low-resource languages such as Urdu. The scarcity of language resources requires effective feature engineering and selection techniques, and machine learning approaches can be effectively applied to Urdu fake news classification. The study presented a novel feature selection measure (NES) to select discriminative features for the Urdu fake news classification. Our proposed method ranked the discriminating terms for filtering, decreasing the data dimensionality and improving the classification performance, as evident from the experimental results. The binary and TF-IDF weighting metrics performed similarly, and we used them as our baseline. To compare and analyze the performance of our feature selection model, we compared it with seven well-known feature selection methods.

Moreover, we evaluated the performance of our proposed model on the BET and UFN datasets. Our analysis of both datasets showed that our proposed approach works well in finding highly discriminative features to classify the Urdu news. Our proposed approach achieved an accuracy of 88% and 90% on the BET and UFN datasets, respectively. In the future, the proposed feature selection model can also be used for other text classification problems. Moreover, we plan to work on language resources to facilitate downstream tasks for the Urdu language.

### Funding

This work is funded by FCT/MEC through national funds and co-funded by FEDER—PT2020 partnership agreement under the project UIDB/50008/2020. The funders had no role in study design, data collection and analysis, decision to publish, or preparation of the manuscript.

### Grant Disclosures

The following grant information was disclosed by the authors:
FCT/MEC through national funds.
FEDER—PT2020 partnership agreement under the project UIDB/50008/2020.

### Competing Interests

Ivan Miguel Pires is an Academic Editor for PeerJ Computer Science.

## Author Contributions

- Muhammad Wasim conceived and designed the experiments, performed the experiments, analyzed the data, performed the computation work, prepared figures and/or tables, authored or reviewed drafts of the article, and approved the final draft.
- Sehrish Munawar Cheema conceived and designed the experiments, performed the experiments, analyzed the data, performed the computation work, prepared figures and/or tables, authored or reviewed drafts of the article, and approved the final draft.
- Ivan Miguel Pires conceived and designed the experiments, performed the experiments, analyzed the data, performed the computation work, prepared figures and/or tables, authored or reviewed drafts of the article, and approved the final draft.

## Data Availability

The code is available at GitHub and Zenodo:

- https://github.com/dr-m-wasim/UrduFakeNewsFS.

- Muhammad Wasim, Sehrish Munawar Cheema, & Ivan Miguel Pires. (2023). Normalized Effect Size (NES): a novel feature selection model for Urdu fake news classification. https://doi.org/10.5281/zenodo.8320957.

The BET Dataset is available at GitHub: https://github.com/MaazAmjad/Datasets-for-Urdu-news.

Institution: Natural Language and Text Processing Laboratory, Center for Computing Research (CIC), Instituto Politécnico Nacional (IPN), Ciudad de México (Mexico City), Mexico

Contact: Maaz Amjad (maazamjad@phystech.edu)

The UFN Dataset is available at GitHub: https://github.com/pervezbcs/Urdu-Fake-News.

Institution: Department of Humanities and Basic Sciences, MCS, National University of Sciences and Technology, Islamabad, Pakistan

Contact: Farkhanda Afzal: (farkhanda@scm.edu.pk)

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
