# Peer review of "Normalized effect size (NES): a novel feature selection model for Urdu fake news classification"

_PeerJ Computer Science, doi:10.7717/peerj-cs.1612_

## Round 0.1 · original submission · Minor Revisions

Please follow review comments.

**Language Note:** The review process has identified that the English language must be improved. PeerJ can provide language editing services - please contact us at copyediting@peerj.com for pricing (be sure to provide your manuscript number and title). Alternatively, you should make your own arrangements to improve the language quality and provide details in your response letter. – PeerJ Staff

Reviewer 1 ·

Basic reporting

The article is in clear English. The text is technically correct. The article conforms to professional standards of courtesy and expression.
The article includes sufficient relevant literature.
The structure of the article conforms to an acceptable format. Figures are relevant to the content of the article.
The submission is ‘self-contained’.
Formal results include clear definitions.

Experimental design

The research is within Aims and Scope of the journal.
The research question is well defined, relevant .
The investigation has been conducted rigorously and to a high technical standard.
The methods are described with sufficient detail.

Validity of the findings

An novel a approach is proposed.
The examples include software validation and verification, accuracy.
The data on which the conclusions are based are provided and are statistically sound.
The conclusions are appropriately stated and connected to the original question investigated,

Additional comments

State clearly the advantage of the proposed approach in the Abstract.
State clearly the novelty of your approach in the Abstract.
Give priority to description of your novel approach and less attention to description of the problem itself in the Abstract.
State clearly in the Conclusion whether you have achieved your goal.
Give the obtained accuracy data in the Conclusion.

Reviewer 2 ·

Basic reporting

The “Abstract” Section is too long and should be re-written due to the difficult of catching the innovation point. It should be approx. 200-250 words
Some Equations in the paper are shown incompletely. To improve the readability of the paper, please explain each variable meaning.
Describe the value of hyperparameters used in experiments
Provide equation numbers for Precision, Sensitivity and F1 Score and all the equations clearly.
Provide mathematical explanations to find the parameters Prediction Time, Classification Error, Model Loss, Precision, Sensitivity and F1 score.
justify differentiating between binary and multiclass classification.
Interpretation of the model is needed. why it was selected, how it was selected?
grammar correction needed.
Proper captioning is needed in figure.
There are many spelling errors in the manuscript. A thoroughly spelling check is required in the revision.
following research article could be referenced in your article: https://doi.org/10.3390/electronics11244096

Experimental design

Need to be improved

Validity of the findings

satisfactory.

---

## Round 0.2 · accepted · Accept

the authors have addressed all of the reviewers' comments.